# Incarceration Exposure and Barriers to Prenatal Care in the United States: Findings from the Pregnancy Risk Assessment Monitoring System

**DOI:** 10.3390/ijerph17197331

**Published:** 2020-10-08

**Authors:** Alexander Testa, Dylan B. Jackson

**Affiliations:** 1Department of Criminology & Criminal Justice, University of Texas at San Antonio, San Antonio, TX 78207, USA; 2Department of Population, Family, and Reproductive Health, Johns Hopkins University, Baltimore, MD 21205, USA; dylan.jackson@jhu.edu

**Keywords:** incarceration, pregnancy, maternal health, prenatal care, health

## Abstract

Previous research demonstrates that exposure to incarceration during pregnancy – either personally or vicariously through a partner – worsens parental care. However, little is known about the specific barriers to parental care that are associated with incarceration exposure. Using data from the Pregnancy Risk Assessment Monitoring System (years 2009–2016), the current study examines the relationship between exposure to incarceration during pregnancy and barriers to prenatal care in the United States. Negative binomial and logistic regression models were used to assess the association between the recent incarceration of a woman or her partner (i.e., incarceration that occurred in the 12 months prior to the focal birth) and several barriers to prenatal care. Findings indicate that exposure to incarceration, either personally or vicariously through a partner, increases the overall number of barriers to prenatal care and this association operates through several specific barriers including a lack of transportation to doctor’s appointments, having difficulty finding someone to take care of her children, being too busy, keeping pregnancy a secret, and a woman not knowing she was pregnant. Policies designed to help incarceration exposed women overcome these barriers can potentially yield benefits for enhancing access to parental care.

## 1. Introduction

A growing body of research demonstrates that incarceration is a stressful life event that significantly increases hardships [1,2] and adversely impacts the health of both incarcerated individuals and their family members [3,4,5,6]. Findings from a growing body of literature also suggest that women who experience incarceration during pregnancy (either personally or vicariously through a partner) have worse infant health outcomes [7,8,9,10,11,12,13]. Moreover, two previous studies from the United States using data from the Pregnancy Risk Assessment Monitoring System (PRAMS) have found that women who report incarceration of either themselves or their partner in the year before birth were more likely to have inadequate prenatal care [12,14]. 

Deficient prenatal care during pregnancy is associated with several infant health problems, including low birth weight and preterm birth [15,16], as well as infant mortality [17,18]. Accordingly, prenatal care is a recommended component of a healthy pregnancy by leading organizations including the American Academy of Pediatrics [19] and the World Health Organization [20]. Furthermore, the Healthy People 2020 initiative includes increasing the proportion of pregnant women who receive early and adequate prenatal care as a key objective in order to reduce the risk of adverse maternal and infant health outcomes [21]. Even so, certain disadvantaged segments of the population continue to experience diminished access to prenatal care [22]. 

Although prior research has established a link between incarceration exposure and inadequacies in prenatal care [12,14], less is known about why pregnant women who encounter the carceral system experience worse prenatal care. Notably, there are unique features of incarceration that make it distinct from other challenging life events that may occur during pregnancy [12,23]. Specifically, incarceration is often a sudden and unexpected life event that forcibly removes a person (i.e., a pregnant woman or her partner) from a household and sustains that removal for an extended period. In this way, incarceration is a uniquely disruptive experience as it often occurs without notice and forcibly restructures households in a way that is akin to major life events such as divorce or death [24]. This sudden shock can be particularly harmful when it occurs during pregnancy by contributing to significant stress, economic hardship, and a series of logistical barriers that can inhibit a mother’s access to essential health care services. 

Notably, there are a few reasons to expect that incarceration exposure may constitute a major life event that is uniquely associated with barriers to receiving prenatal care. First, incarceration is a destabilizing event that can exacerbate the economic hardship of a household [1,2]. Prior work finds that the incarceration of a male partner increases economic strain on their female partners, leading to an increased likelihood of these women having to work multiple jobs [25], enroll in public assistance programs [26], and rely on financial support from family members [1]. In turn, this may generate barriers related to the costs of prenatal care. Second, having an incarcerated partner can also increase the burden of performing essential services, such as childcare and daily household activities, on the remaining household members [14,27,28,29]. Thus, the incarceration of a partner can result in greater time restrictions and challenges with obtaining childcare, which can serve as barriers to the receipt of routine prenatal care. Third, while incarcerated, a person can lose their health insurance and may need to re-establish health care upon release [30]. Such a disruption to health insurance can be a barrier to receiving adequate and timely prenatal care. To be sure, formerly incarcerated populations are found to have lower rates of health insurance and use various health care services less often compared to the general population [31,32]. Fourth, past research suggests incarceration is negatively associated with the ownership of a vehicle [33] and incarceration exposed populations often report difficulties in accessing reliable transportation [2,34,35], thereby creating potential transportation barriers to accessing prenatal care. To be sure, adequate access to transportation services is, in many cases, a necessary component of accessing prenatal care [36,37]. Fifth, recent work also links incarceration to decreased trust in medical authorities [38], possibly leading to an avoidance of medical institutions more generally among justice-involved populations [39]. To the extent that this occurs among pregnant women, the desire for prenatal care, as well as efforts to seek health care services at the earliest signs of pregnancy may be reduced. Finally, because incarceration tends to be associated with relationship distrust between partners and relationship dissolution [40,41,42], pregnant women may be more likely to avoid prenatal care (particularly during the early stages of pregnancy) because they are keeping a pregnancy a secret from their partner. 

In summary, prior research has documented that women exposed to incarceration during pregnancy report less access to prenatal care [12,14]. Moreover, the above-mentioned literature suggests that incarceration can be a pivotal life event that generates challenges in accessing timely and adequate prenatal care. Yet, research has not explored the connection between incarceration exposure and reports of barriers to prenatal care during pregnancy. Accordingly, the purpose of this study is to expand existing knowledge on incarceration exposure and prenatal care by investigating the specific barriers to prenatal care faced by incarceration exposed women. Accordingly, our objective in conducting this investigation is to produce findings that can (1) aid in the identification of modifiable risk factors that pose barriers to prenatal care and (2) point to clear intervention opportunities, which in turn can lead to expanded prenatal care access among this vulnerable population. Specifically, this work addresses two main questions:Do women exposed to incarceration during pregnancy face more barriers to prenatal care than women without incarceration exposure?Are there specific barriers to which incarceration exposed women are particularly prone?

## 2. Materials and Methods 

Data for this study are from the Pregnancy Risk Assessment Monitoring System. PRAMS is an ongoing population surveillance system in the United States conducted by the Centers for Disease Control and Prevention (CDC) in conjunction with state health departments. Using birth certificate records, participating states conduct an annual stratified sample of approximately 1000 to 3000 women who are state residents and recently delivered a live birth. Sample sizes are determined according to a stratification plan, number of births, and available budgets. Retrospective data are collected from recent mothers via a questionnaire that is linked with birth certificate data. Women are mailed a survey approximately 2–6 months following birth. Up to five contact attempts are made by mail and states follow up with non-responders by phone call within a week of the last mailed survey. 

During the study period (2009–2016), there were approximately 4 million births per year in the United States [43]. The PRAMS sampling strategy (based on birth certificate data) is meant to generate a sample representative of all women who delivered a live-born infant. To do so, the analysis relies on survey weights which include three components: the sampling weight, a nonresponse adjustment, and a noncoverage adjustment. Specifically, since birth certificate data are available for both responders and non-responders, available information from non-responders is used to adjust for a non-response. The birth certificate file available from each state is also compared with the PRAMS sampling frame to identify eligible records that were not included in the survey sample and adjust for non-coverage. Because some states do not participate, the PRAMS survey currently covers approximately 83% of live births within the United States [39]. Furthermore, since certain states do not ask questions about specific experiences such as incarceration exposure and barriers to prenatal care, the current study was limited to surveys administered in 34 states and New York City that included questions on all measures included in this study (see Appendix A). More information on the PRAMS survey and methodology can be found in Shulman et al [44]. The use of PRAMS data for the current study were approved by the CDC.

### 2.1. Dependent Variable

Women who answered a survey question indicating that they did not receive prenatal care as early as they wanted or did not go for prenatal care were asked a series of follow-up questions regarding the specific barriers that prevented them from getting timely prenatal care. Women answered either true or false to 11 questions regarding barriers to prenatal care listed in Appendix B. Reponses on the items were summed into a prenatal care barriers index, which ranges from 0–11 (Cronbach’s Alpha = 0.761). Overall, these 11 items encompass a wide range of barriers broadly defined, which encompass structural, cognitive, and knowledge-based issues that have been found to inhibit women from getting adequate prenatal care in prior literature [45,46,47]. 

### 2.2. Independent Variable

Incarceration exposure was self-reported by recent mothers. Specifically, incarceration exposure was measured using a survey item that asks women whether in the 12 months before the current birth “I or my husband/partner went to jail” (1 = yes, 0 = no). Accordingly, this item measured whether a mother or her partner was incarcerated during pregnancy or shortly before the pregnancy began. While an affirmative response could indicate the incarceration of either the mother or her husband/partner, because over 90% of those incarcerated in prisons [48] and approximately 85% of those incarcerated in jails [49] in the United States are male, in most cases, the incarceration likely refers to the husband/partner. 

### 2.3. Control Variables 

Control variables measured several sociodemographic characteristics including maternal age (17 or younger [reference], 18–24, 25–29, 30–34, 35 or older), maternal race/ethnicity (White [reference], Hispanic, Black, other race/ethnicity), whether the mother was a college graduate (1 = 16 or more years of education, 0 = less than 16 years), currently married (1 = yes, 0 = no), number of prior births (0 [reference], 1, 2, 3+), whether the current pregnancy was planned (1 = yes, 0 = no), and income levels (<$10,000 [reference], $10,000–14,999, $15,000–19,999, $20,000–24,999, $25,000–34,999, $35,000–$49,999, $50,000 or greater). Models included binary indicators for state of residence and year of birth to account for variation in barriers to prenatal care across place and time. 

### 2.4. Analytic Approach

The analysis proceeded in a series of stages. First, the summary statistics of the analytic sample stratified by incarceration are presented. We then applied a difference in means *t*-tests to assess which variables significantly differed between incarceration-exposed and non-incarceration-exposed women. Second, we examined the association between incarceration exposure and the total number of barriers (range = 0–11) in the full sample of women (N = 194,600). Negative binomial regression was used given the outcome variable is positively skewed with a large number of zero values (i.e., most women reported not experiencing barriers to prenatal care). Third, we then limited the analysis to the subsample of women who reported not receiving prenatal care as early as they wanted or not receiving prenatal care (N = 34,658) and assessed how incarceration exposure was related to the 11 specific barriers to prenatal care among this subset of women. Given the binary nature of these 11 outcome variables, we estimated separate logistic regression models using responses to each of the 11 barriers to prenatal care as the dependent variables, while controlling for the set of covariates. All statistical analyses were conducted using Stata V.15.1. Models were adjusted for survey weights and strata information to account for the complex design of the PRAMS survey. Missing data were addressed using multiple imputation with chained equations, resulting in the utilization of 20 multiply imputed data sets. 

## 3. Results

Summary statistics are reported in Table 1. Approximately 4.7% of the sample reported incarceration exposure. Incarceration exposed women experienced a greater number of barriers to prenatal care (0.915 vs. 0.363, *p* < 0.001). Incarceration exposed women were also younger, were more likely to be Black or Hispanic, were less likely to be college graduates, were less likely to be married, had a higher number of prior births, were less likely to report that the most recent pregnancy was planned, and had lower levels of income. The prevalence of specific barriers to prenatal care stratified by incarceration exposure are reported in Figure 1. Incarceration exposed women reported a higher prevalence of several barriers including not having enough money (38.3% vs. 31.7%, *p* < 0.001), not having transportation to the doctor’s office or clinic (26.8% vs. 11.7%, *p* < 0.001), not having a Medicaid card (37.5% vs. 29.1%, *p* < 0.001), being too busy (31.1% vs. 16.5%, *p* < 0.001), having difficulty finding someone to take care of her children (13.5% vs. 7.8%, *p* < 0.001), keeping pregnancy a secret (22.4% vs. 12.6%, *p* < 0.001), not knowing they were pregnant (49.4% vs. 35.7%, *p* < 0.001), and not wanting prenatal care (5.8% vs. 3.9%, *p* = 0.035).

Table 2 presents the results of the negative binomial regression of the number of prenatal care barriers on incarceration exposure. The bivariate results shown in Model 1 demonstrate that incarceration exposure was associated with an approximately 2.5 times greater rate of barriers to prenatal care (IRR = 2.484, 95% CI = 2.267, 2.721). After including the control variables in Model 2, women exposed to incarceration were expected to encounter barriers to prenatal care at an approximately 1.6 times greater rate (IRR = 1.558, 95% CI = 1.416, 1.714). The marginal effects of the coefficients are reported visually in Appendix C. As the appendix displays, the impact of incarceration on the number of barriers to incarceration serves as one of the strongest effect sizes, apart from planned pregnancy and the highest levels of income. 

Next, we estimated whether incarceration elevated the odds of experiencing specific barriers to prenatal care while controlling for covariates. The results of the series of logistic regression models are illustrated in Figure 2. Incarceration exposure elevated the odds of several specific barriers to prenatal care, including lacking transportation to get to the clinic or doctor’s office (OR = 1.795, 95% CI = 1.470, 2.192), having too many other things going on (OR = 1.731, 95% CI = 1.453, 2.062), having no one to take care of children (OR = 1.413, 95% CI = 1.090, 1.831), keeping pregnancy a secret (OR = 1.404, 95% CI = 1.145, 1.722), and not knowing they were pregnant (OR = 1.396, 95% CI = 1.189, 1.639).

## 4. Discussion

The current study aimed to extend prior literature on incarceration and prenatal care access [12,14] by assessing whether incarceration exposure during pregnancy contributed to a greater number of prenatal care barriers, as well as which specific barriers to prenatal care were elevated among incarceration exposed women. The study yielded two key findings. First, women exposed to incarceration during pregnancy face significantly more barriers to prenatal care relative to women who were not exposed to incarceration. Second, incarceration is associated with independent increases in the odds of certain barriers to prenatal care, including having a lack of transportation to get to a doctor, having no one to care for children, being too busy to get prenatal care, as well as lacking adequate prenatal care because of keeping pregnancy a secret or not knowing they were pregnant. Below we discuss why incarceration may influence these specific barriers to prenatal care and some possible interventions that could help alleviate these barriers and expand prenatal care access among incarceration exposed women. 

Many of these specific barriers can be understood within the broader literature of the consequences of incarceration, suggesting that ameliorating the hardship and logistical challenges so often generated by incarceration could improve prenatal care access among incarceration exposed women. Consistent with past research, study findings highlighted a lack of transportation as a hardship stemming from exposure to incarceration [2,34,35]. Past research has shown that incarceration is associated with a reduction in ownership of household assets such as a vehicle [31] and exacerbates financial hardship [1,2]. Therefore, incarceration exposed populations may need to rely upon public transportation to access health care services [2]. Given the pattern of economic hardship and lower rates of automobile ownership that has emerged in prior research, one potential intervention is to expand access to low-cost public transportation to increase accessibility to prenatal care among incarceration exposed women. For example, this can be achieved through providing vouchers to taxis or ride-sharing services (i.e., Uber; Lyft), as well as public transportation (i.e., subway, bus passes). Indeed, prior research has found that transportation incentives can increase prenatal care compliance [50]. Second, incarceration exposure was also associated with having difficulty finding someone to take care of children and being too busy as barriers to prenatal care. This is not particularly surprising within the context of the extant literature, given that incarceration exposure is associated with women having to take on additional employment [1] as well as bearing a greater responsibility for childcare and essential household activities [14,27,28,29]. Incarceration may present unique challenges for women, given that incarceration exposed mothers are more likely than fathers to report having lived with/taken care of their child prior to incarceration [51]. These findings from prior literature highlight that time constraints are a rarely considered consequence of incarceration that can generate significant social and health inequities, including barriers to essential health services such as prenatal care. Considering these findings, the provision of targeted resources to incarceration exposed women, such as vouchers for childcare services or providing onsite childcare services at prenatal care clinics may be beneficial approaches to minimize barriers to prenatal care [52]. Likewise, given the logistical barriers noted above, incarceration exposed women may benefit from expanded telemedicine, which can provide more flexible appointments [53] and enhance accessibility for women who face time constraints, childcare responsibilities, and difficulty with reliable transportation [54]. 

Finally, barriers to prenatal care such as keeping a pregnancy secret or being unaware of the pregnancy were also elevated among incarceration exposed women. Incarceration is a well-documented strain on romantic partnerships [55], as well as social relationships more generally [56]. Moreover, prior research finds that incarceration exposed populations document patterns of jealousy and suspicions pertaining to sexual infidelity of their partners [40]. Therefore, it is possible that women may be keeping a pregnancy a secret in order to reduce strife between themselves and their partners, as well as with family members who they may rely on for support [1]. Finally, incarceration exposed women were less likely to receive prenatal care because of not knowing they were pregnant. To the extent that incarceration exposed women have lower levels of gestational and health literacy [57,58,59,60], these women may be more prone to not receiving adequate prenatal care (particularly during the first trimester) because of lacking knowledge about the early signs of pregnancy. Furthermore, these barriers (keeping pregnancy a secret and not knowing they were pregnant) could be linked to an unintended pregnancy or ambivalence about being pregnant [61]. In this case, the best course of action may be expanding education and information to segments of the population at high risk for incarceration exposure (i.e., low socioeconomic status women or those living in communities with high rates of incarceration) to help women better plan for pregnancy and obtain the necessary health care services. As Epstein and colleagues [61] suggest, “in order to improve rates of early initiation of prenatal care, programs and health care providers will need to address factors that affect women before they become pregnant, including education and services to help women plan and time their pregnancies.” These forms of education can be provided in correctional facilities, as well as through community supervision agencies (i.e., probation and parole) to extend important information to justice involved populations.

Broadly speaking, the results of this study also suggest that more direct efforts should be made to expand prenatal care access among incarceration exposed women. One possibility is to utilize health care workers to target incarceration exposed women as a means of providing both education and resources to enhance prenatal care. Indeed, as Dumont and colleagues [14] have previously suggested, “incarceration history marks a high-risk population that may benefit from intensified attention by healthcare workers to improve their rates of prenatal care.” A useful approach could be to coordinate interventions between correctional facilities and public health agencies to target visiting hours as a means of providing information about prenatal care to pregnant women. Alternatively, resources can be provided to expand the availability of home visiting programs to incarceration exposed women. Typically, such programs provide trained health professionals to visit the homes of vulnerable women both during pregnancy and following birth to provide information and resources (i.e., transportation, health, social services, and basic necessities) in order to ensure maternal and infant wellbeing [62]. Prior investigations have found that participation in home visiting programs are associated with increases in the amount of prenatal care visits [63]. The implementation of such a program may also provide the collateral benefits of helping women to identify and overcome some key barriers to prenatal care. One useful direction for future research would be to investigate the moderating role of state level policies or features on the association between incarceration exposure and barriers to prenatal care. Doing so may begin to illuminate how concrete legislation in individual states influences prenatal care access among incarceration exposed women. 

There are a few limitations in the current study that can be expanded upon in future research. First, we cannot disentangle whether the incarceration was experienced by the recent mother or her partner. Because most persons incarcerated in the United States are male, it is likely that in most of these cases, the responses refer to a partner’s incarceration [48,49]. Nevertheless, barriers to prenatal care may differ based on who experienced the incarceration. For instance, some research does find that for very high-risk women, serving time incarcerated may improve infant health outcomes [7,64]. This suggests the possibility that being incarcerated potentially benefits pregnant women with lower resources and means. Accordingly, a fruitful line of inquiry would be to examine whether incarceration can improve access to prenatal care for some at-risk women. For instance, prior research has suggested that “incarceration may constitute a period of relative stability and improved access to prenatal care that can improve birth outcomes” [65]. Relatedly, some of the barriers included in the current study may be less relevant (i.e., lacking transportation) or operate differently (i.e., could not get an appointment; did not have money or insurance) depending on whether a mother is incarcerated for the duration of her pregnancy. Ideally, future research could explore alternative groupings of incarceration exposure to generate a deeper understanding of this issue. For instance, it would be useful for research to examine prenatal care barriers depending on whether a woman or her husband/partner was the person incarcerated and whether that incarceration occurred for only some or all the pregnancy. 

Second, the PRAMS survey asks about incarceration in jail. However, this measure is used as a proxy for incarceration in any facility given that jail and prison are both used interchangeably among the general public [12,14,23]. Third, the question asks about whether incarceration was experienced at some point in the 12 months prior to the current birth. Accordingly, we cannot ascertain when during the pregnancy (i.e., shortly before, 1^st^, 2^nd^, 3^rd^ trimester) the incarceration event occurred. It would be useful for future research to investigate the association between the timing of incarceration and barriers to prenatal care during pregnancy. Fourth, the incarceration measure only differentiates those who experienced incarceration, but does not capture features such as the length of the sentence. Even so, prior research on the consequences of incarceration for health tend to find that the effects of the length of incarceration on health outcomes are less important than incarceration itself [66,67]. Still, future work exploring the association between the length of an incarceration sentence and barriers to prenatal care would be useful. Fifth, there may be relevant variables that could not be captured in the current study, such as the accessibility to health care providers within local areas, ownership of an automobile, or the age of any children currently living with the mother. One possible avenue for future research is to investigate how the availability of resources and household composition might alter the relationship between incarceration exposure and barriers to prenatal care. Relatedly, while we included a robust set of covariates assessing factors related to both incarceration and barriers to prenatal care, future research should examine measures that capture other disadvantaged groups, such as drug users [68] or homeless populations [52,69]. Finally, the PRAMS survey only asked about barriers to prenatal care among those who reported issues related to the adequacy of prenatal care. Accordingly, we could not examine whether incarceration exposed women experienced worse prenatal care because of having more barriers to prenatal care [70]. 

## 5. Conclusions

The findings from the current study reaffirm the challenges that incarceration exposed women face in obtaining adequate prenatal care, as well as extend previous research on this topic by identifying the specific barriers to prenatal care that are elevated in the face of incarceration exposure. Specifically, the findings highlight that women exposed to incarceration during pregnancy confront a greater overall number of barriers to prenatal care and incarceration exposure is associated with an increased risk of a specific set of barriers to prenatal care. These findings are useful for informing prevention and intervention efforts aimed at providing services that diminish these barriers to prenatal care among pregnant women whose lives have been adversely impacted by the criminal justice system. 

## Figures and Tables

**Figure 1 ijerph-17-07331-f001:**
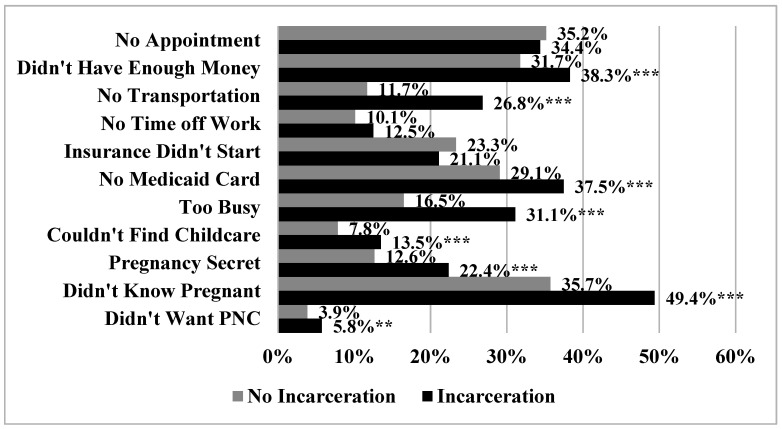
Prevalence of Specific Barriers to Prenatal Care Stratified by Incarceration (*N* = 34,658). *** *p* < 0.001, ** *p* < 0.01.

**Figure 2 ijerph-17-07331-f002:**
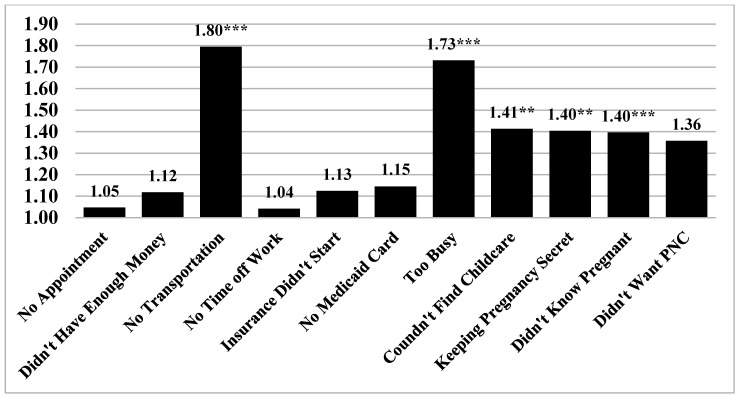
Results of Logistic Regression of Incarceration Exposure on Individual Prenatal Care Barriers (N = 34,658). *Notes*: Numbers reported in the bar graph represent adjusted odds ratios; All models control for covariates reported in Table 2. *** *p* < 0.001, ** *p* < 0.01.

**Table 1 ijerph-17-07331-t001:** Weighted Summary Statistics Stratified by Incarceration Exposure.

Variables	No Incarceration (N = 185,406)	Incarceration (N = 9194)	*p*-Value
Total Barriers	0.363	0.915	<0.001
*Maternal Age*			
17 or Younger	2.3%	4.7%	<0.001
18–24	26.2%	50.9%	<0.001
25–29	29.4%	26.0%	<0.001
30–34	26.9%	12.7%	<0.001
35+	15.2%	5.7%	<0.001
*Maternal Race/Ethnicity*			
White	59.0%	48.4%	<0.001
Hispanic	13.3%	27.8%	<0.001
Black	19.8%	17.8%	0.027
Other Race/Ethnicity	7.9%	6.0%	<0.001
College Graduate	32.8%	5.7%	<0.001
Currently Married	62.0%	21.0%	<0.001
*Number of Prior Births*			
0	40.3%	41.0%	0.432
1	32.5%	27.9%	<0.001
2	16.2%	16.9%	0.379
3+	10.9%	14.1%	<0.001
Pregnancy Planned	50.7%	26.5%	<0.001
*Income Levels*			
Less than $10,000	21.7%	56.1%	<0.001
$10,000–$14,999	5.5%	9.3%	<0.001
$15,000–$19,999	3.6%	4.7%	0.016
$20,000–$24,999	10.4%	13.2%	<0.001
$25,000–$34,999	10.9%	8.0%	<0.001
$35,000–$49,999	10.5%	4.1%	<0.001
$50,000 or Greater	37.4%	4.7%	<0.001

**Table 2 ijerph-17-07331-t002:** Results of Negative Binomial Regression of Incarceration Exposure on Number of Prenatal Care Barriers (*N* = 194,600).

Variables	Model 1		Model 2	
	IRR	95% CI	IRR	95% CI
Incarceration Exposure	2.484 ***	(2.267, 2.721)	1.558 ***	(1.416, 1.714)
*Maternal Age*				
17 or Younger (Reference)	-	-	-	-
18–24			0.904 ^†^	(0.804, 1.016)
25–29			0.788 ***	(0.695, 0.893)
30–34			0.695 ***	(0.608, 0.795)
35+			0.687 ***	(0.595, 0.794)
*Maternal Race/Ethnicity*				
White (Reference)	-	-	-	-
Hispanic			1.191 ***	(1.108, 1.281)
Black			1.103 **	(1.032, 1.180)
Other Race/Ethnicity			1.552 ***	(1.432, 1.683)
College Graduate			0.867 ***	(0.806, 0.932)
Currently Married			0.848 ***	(0.797, 0.901)
*Number of Prior Births*				
0 (Reference)	-	-	-	-
1			0.949 ^†^	(0.896, 1.006)
2			1.017	(0.947, 1.092)
3+			1.263 ***	(1.153, 1.384)
Pregnancy Planned			0.523 ***	(0.495, 0.552)
*Income Levels*				
Less than $10,000 (Reference)	-	-	-	-
$10,000–$14,999			1.020	(0.923, 1.129)
$15,000–$19,999			0.897 ^†^	(0.796, 1.012)
$20,000–$24,999			0.837 ***	(0.770, 0.910)
$25,000–$34,999			0.733 ***	(0.672, 0.799)
$35,000–$49,999			0.611 ***	(0.553, 0.676)
$50,000 or Greater			0.324 ***	(0.296, 0.356)
Constant	0.535 ***	(0.468, 0.612)	1.093	(0.910, 1.313)
State Dummy Variables	Yes		Yes	
Year Dummy Variables	Yes		Yes	

*** *p* < 0.001, ** *p* < 0.01, ^†^
*p* < 0.10.

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
