# Peer review of "Incarceration Exposure and Barriers to Prenatal Care in the United States: Findings from the Pregnancy Risk Assessment Monitoring System"

_ijerph, 2020, doi:10.3390/ijerph17197331_

Round 1

Reviewer 1 Report

Generally speaking, submission is interesting even if results are clearly expected, so that a paragraph devoted to policy interventions should be interesting for the readers, since the fact that incarceration is a barrier to prenatal care is obvious, what to do to avoid the problem is less obvious. Please comment.

General comment: please put the submission in past tense. The study is already ended and past tense is more correct. All the submission should be shortened to make the reading more fluent and easy

Abstract:

Pregnancy Risk Assessment Monitoring System (PRAMS) from which Country? Clarify in the abstract to better understand the context

 lack of transportation to doctor’s appointment. Transportation? Please use another term

Introduction:

Introduction is too lengthy. Shorten and Please focus on the novelties of the present submission and discuss if they had been treated in previous investigations, if not why not and what this study brings as new message.

Methods:

1400-3000 women, which percentage of total incarcerated pregnant women in US?

No mention to applied statistic tests? Please amend and explain the meaning of each test, why it was applied

Figure 1 is confused , numbers are too big, lines too narrow, please modify.

Figure 2 is messy too. Please make numbers smaller

Author Response

  1. Generally speaking, submission is interesting even if results are clearly expected, so that a paragraph devoted to policy interventions should be interesting for the readers, since the fact that incarceration is a barrier to prenatal care is obvious, what to do to avoid the problem is less obvious. Please comment.

Response: We thank the reviewer for the interest in the manuscript. We have taken care to revise the manuscript in accordance with the reviewer’s suggestions. In addition, we have added to the discussion section policy interventions that may be of interest to the reader.

  1. General comment: please put the submission in past tense. The study is already ended and past tense is more correct. All the submission should be shortened to make the reading more fluent and easy

Response: We have revised the manuscript to now be in past tense.

  1. Pregnancy Risk Assessment Monitoring System (PRAMS) from which Country? Clarify in the abstract to better understand the context

Response: We have now clarified in the abstract and under the materials and methods section that the PRAMS data are from the United States.

  1. lack of transportation to doctor’s appointment. Transportation? Please use another term

Response: We have now sought to better clarify in the manuscript this point. Specifically, we opted to retain the term “transportation” because this is the terminology used in the PRAMS survey questionnaire which reads “I had no transportation to get to the clinic or doctor’s office.” Furthermore, the phrasing “lack of transportation” is also consistent with prior research using PRAMS data (Davis et al., 2004; Epstein et al., 2009), as well as other research on the topic using different data sources (Braveman et al., 2000). Even so, we have aimed to clarify this when discussing the transportation variable.

Braveman, P., Marchi, K., Egerter, S., Pearl, M., & Neuhaus, J. (2000). Barriers to timely prenatal care among women with insurance: the importance of prepregnancy factors. Obstetrics & Gynecology95(6), 874-880.

Davis, K., Baksh, L., Bloebaum, L., Streeter, N., & Rolfs, B. (2004). Barriers to adequate prenatal care in Utah. PRAMS Perspectives1(1), 1-8.

Epstein, B., Grant, T., Schiff, M., & Kasehagen, L. (2009). Does rural residence affect access to prenatal care in Oregon?. The Journal of Rural Health25(2), 150-157.

  1. Introduction is too lengthy. Shorten and Please focus on the novelties of the present submission and discuss if they had been treated in previous investigations, if not why not and what this study brings as new message.

Response: We have revised the introduction to shorten it and focus more on the novelty of the current study.

  1. 1400-3000 women, which percentage of total incarcerated pregnant women in US?

Response: The number of births in the United States fluctuate by year and differ by state due to differences in population size However, During the study period (2009-2016), there were approximately 4-million births per year in the United States (Hamilton et al., 2019). While the PRAMS data do not capture the population of all births, the sample, which is based on birth certificate records are used in each participating jurisdiction is meant to generate a sample representative of all women who delivered a live-born infant. Furthermore, we now note that Sample sizes are determined according to stratification plan, number of births, and available budgets (Shulman et al., 2018). Finally, the analysis weights include 3 components: the sampling weight, a nonresponse adjustment, and a noncoverage adjustment. Because birth certificate data are available for both responders and nonresponders, the information available on nonresponders can be used to adjust for nonresponse and to understand factors associated with survey nonresponse. The final cumulative birth certificate file from each state is compared with the PRAMS sampling frame to identify eligible records that were missed and compute noncoverage adjustments. We have revised the section introducing the data to now include these details explained above.

Hamilton, B. E., Martin, J. A., Osterman, M. J., Driscoll, A. K., & Rossen, L. M. (2018). Births: provisional data for 2017 (Vital Statistics Rapid Release Report 004). Hyattsville, MD: National Center for Health Statistics.

Shulman, H. B., D’Angelo, D. V., Harrison, L., Smith, R. A., & Warner, L. (2018). The pregnancy risk assessment monitoring system (PRAMS): overview of design and methodology. American Journal of Public Health108(10), 1305-1313.

  1. No mention to applied statistic tests? Please amend and explain the meaning of each test, why it was applied

Response: We have revised the section describing the method to now carefully describe the meaning of each test and discuss why it was applied.

  1. Figure 1 is confused , numbers are too big, lines too narrow, please modify.

Response: We have now revised the format of Figure 1, by widening the lines and making the numbers smaller to improve readability.

  1. Figure 2 is messy too. Please make numbers smaller

Response: We have revised figure 2 and made the numbers smaller to improve readability.

Reviewer 2 Report

The submitted manuscript, “Incarceration exposure and barriers o prenatal care: Findings from the Pregnancy Risk Assessment Monitoring System”, suggests that pregnant women whose partner or themselves were incarcerated experience more barriers to prenatal care relative to women without incarceration exposure. While the results are of interest, and the discussion does not exceed the reported findings, there are several issues with the current version of this manuscript that need to be addressed prior to it being accepted for publication. These concerns are listed below. My biggest concern (point 1) is that I am uncertain as to whether these barriers are unique to women with incarceration exposure. The authors need to develop a better argument for why the listed barriers (and to a lesser extent the proposed solutions) are unique to the population under study.

  • How is incarceration exposure’s influence on access to prenatal care different for access to medical care issues experienced by other disadvantaged groups?
    • The authors describe their incarcerated group as follows: “younger, more likely to be black or Hispanic, less likely to be college graduates, less likely to be married, had a higher number of prior births, were less likely to report the most recent pregnancy was planned, and had lower levels of income.” For me this is the perfect comparison group of non-incarcerated women. What is the barrier to prenatal care? Incarceration or being younger, more likely to be black or Hispanic, less likely to be college graduates, less likely to be married, had a higher number of prior births, were less likely to report the most recent pregnancy was planned, and had lower levels of income?
    • Comparisons to other disadvantaged groups should also be made, particularly since the presented analyses suggest that being white, educated, married, a planned pregnancy, and an income of $20,000 or more are associated with reducing the risk that incarceration has on accessing prenatal care.
    • Are the effects of incarceration exposure greater (or different) form the effects of other disadvantages on access to prenatal care? For me this is an issue that needs to be addressed in much more detail by the authors.
  • Total number of barriers. Am I correct in assuming the total number of barriers was calculated by summing the women’s responses to the 11-specific barriers? The definition of total barriers needs to be explicitly stated.
    • In the abstract the authors wrote: “… the association between incarceration of a woman or her partner in the year before birth and …”
    • How is incarceration defined?
    • Is the length of incarceration known? If yes, does this make a difference? An unknown length of incarceration should be addressed in the limitations section.
    • In the year before the birth suggestions that the woman/partner could have been incarcerated and then released prior to the woman becoming pregnant. Is the timing of the pregnancy known? If yes, does when during the pregnancy (before, 1st, 2nd, 3rd trimester) make a difference? An unknown timing of pregnancy should be addressed in the limitations section.
    • If both length and timing of incarceration known, does their combined potential effect make a differenc
  • Who is incarcerated? While the authors address this issue in the limitations section, I feel that it is not given enough attention. Can the authors talk about any potential benefits that a young, non-white, poorly educated, woman from a low SES neighborhood might obtain in terms of access to prenatal care if she is incarcerated? I stipulate this population as it describes the present sample. However, can being incarcerated potentially benefit pregnant women with lower resources and means?
  • The authors make a few suggestions (taxi vouchers, public assistance, vouchers for childcare) that might mitigate the effects of incarceration exposure on accessing prenatal care. Have these factors been useful in mitigating prenatal care in other disadvantaged populations? This topic needs to be better developed as to why the authors think they may be beneficial for the population under study above and beyond that observed in other disadvantaged populations.
  • In the Discussion the author write: “As such, this may result in a reduction in ownership of household assets such as a vehicle [31], as well as exacerbate financial hardship, thereby making public transportation costlier and more difficult to obtain.”
    • While semantic in nature, it is incorrect to conclude that incarceration exposure is associated with costlier and more difficult to obtain public transportation. The cost and frequency of public transportation is independent of the women’s incarceration status. Incarceration exposed women may experience greater financial difficulties which in turn may make it financially difficult to use public transportation, but it does not make public transportation costlier.
  • How was access to early childcare defined?
  • Figure 1. It would be useful if the authors identify which barriers differ significantly between the 2 groups.
  • Why is the sample size for Figure 1 (N = 34,658) different from the sample listed for Table 1 (N = 194,600)?
  • Why are “Didn’t want PNC” and “Didn’t know pregnant” listed as barriers to prenatal care? Being too busy, not having childcare, time of work, enough money, a Medicaid care, … cannot be seen as barriers if the person has either stipulated they do not wish to seek prenatal care or are do not know that they require medical care. Please clarify.
    • Do the results of the initial analysis (number of barriers) change if these two “barriers” are removed from the total score of barriers?
  • Figure 2. The labelling for “Didn’t want PNC appears to be incorrect. The bar graph suggests an OR of greater than 1.40.
  • Net of covariates. This is a phrase that I have not come across before. I asked three statisticians how they would interpret the following sentence that authors wrote I the Results section (“To do so, we conduct separate logistic regression models using responses to each of the 11 barriers to prenatal care as the outcome variable, net of covariates.") and received three different responses. One believed that the phrase suggested that the covariates were included in the analyses, one believed that the covariates were excluded from the analyses, and the third did not know whether the covariates were or were not included. I ask for clarification since my interpretation is without covariates. If my interpretation is true, then the text appears to report OR without the influence of the covariates while Figure 2 appears to report ORs after controlling for the covariates.

Note: Significance of Content: I rated this item as low since this was the best option available. This could be rated higher if the authors are able to make the argument that the barriers to prenatal care listed in the study are unique to pregnant women with incarceration exposure. This has not been adequately done in the present version of the manuscript.

Author Response

  1. The submitted manuscript, “Incarceration exposure and barriers o prenatal care: Findings from the Pregnancy Risk Assessment Monitoring System”, suggests that pregnant women whose partner or themselves were incarcerated experience more barriers to prenatal care relative to women without incarceration exposure. While the results are of interest, and the discussion does not exceed the reported findings, there are several issues with the current version of this manuscript that need to be addressed prior to it being accepted for publication. These concerns are listed below. My biggest concern (point 1) is that I am uncertain as to whether these barriers are unique to women with incarceration exposure. The authors need to develop a better argument for why the listed barriers (and to a lesser extent the proposed solutions) are unique to the population under study.

Response: We thank the reviewer for their helpful and thorough comments on the manuscript. As we note below in our responses, we have taken care to address each of these responses. We believe that in doing so, the authors comments have strengthened the quality and overall contribution of the manuscript. 

  1. How is incarceration exposure’s influence on access to prenatal care different for access to medical care issues experienced by other disadvantaged groups?

Response: As we describe in further detail to the specific comments below, we have now altered the text of the manuscript to clarify that findings of the manuscript show that incarceration exposed women face more barrier to prenatal care than non-incarceration exposed women, even when holding other markers of sociodemographic disadvantage (i.e. race, age, marital status, education, income) constant. We have now altered the text to clarify this point. In addition, in the comment below we further detail this finding to clarify our contribution.

  1. The authors describe their incarcerated group as follows: “younger, more likely to be black or Hispanic, less likely to be college graduates, less likely to be married, had a higher number of prior births, were less likely to report the most recent pregnancy was planned, and had lower levels of income.” For me this is the perfect comparison group of non-incarcerated women. What is the barrier to prenatal care? Incarceration or being younger, more likely to be black or Hispanic, less likely to be college graduates, less likely to be married, had a higher number of prior births, were less likely to report the most recent pregnancy was planned, and had lower levels of income?

Response: We thank the reviewer for taking care to notice that the incarceration exposed women have many characteristics that make them at-risk for less prenatal care. Accordingly, we have opted to use multivariate regression analysis which adjusts for these observable factors. Put another way, our reported results show that incarceration exposure remains an independent risk factor for more prenatal care barriers, even when holding age, race, education, and income constant. Moreover, the results reported in Table 2 signify that Black and Hispanic women have slightly elevated risks for more prenatal care barriers compared to White women, and higher levels of education and higher levels of income are negatively associated with prenatal care barriers.

  1. Comparisons to other disadvantaged groups should also be made, particularly since the presented analyses suggest that being white, educated, married, a planned pregnancy, and an income of $20,000 or more are associated with reducing the risk that incarceration has on accessing prenatal care.

Response: As mentioned above, out of a recognition of how certain factors such as race, education, marriage, planned pregnancy, and income levels are related to both prenatal care barriers, as well as the likelihood of incarceration, we have controlled for these factors in a multivariate context. While we believe this is a theoretically relevant and robust set of control variables, we have now added to the limitation section suggesting that there may be other disadvantaged groups for which comparisons should be made based on the existing literature.

  1. Are the effects of incarceration exposure greater (or different) form the effects of other disadvantages on access to prenatal care? For me this is an issue that needs to be addressed in much more detail by the authors.

Response: We thank the reviewer for this helpful comment. Unfortunately, the analytic method used (negative binomial regression) does not provide us with a clear way to standardize the magnitude of the effect across variables. To show the relative magnitude of incarceration compared to other variables we computed the marginal effects of each variable in the model and visualized these in appendix B of the revised manuscript. As the appendix shows, the impact of incarceration is rather large, having a larger overall impact for instance than Hispanic, Black, college graduate, married, prior births, as well as most age and income categories. Even so, planned pregnancy and the highest levels of income (i.e. $50,000 or more) had the largest overall effects on the total number of barriers.

  1. Total number of barriers. Am I correct in assuming the total number of barriers was calculated by summing the women’s responses to the 11-specific barriers? The definition of total barriers needs to be explicitly stated.

Response: The reviewer is correct that the total number of barriers measure is generated by summing the responses (1 = yes; 0 = no) to the 11-specific barriers. We have now clarified this in the manuscript under section 2.1 Dependent Variable.

  1. In the abstract the authors wrote: “… the association between incarceration of a woman or her partner in the year before birth and …”

Response: We have now re-written this portion of the abstract to clarify how the variable is coded.

  1. How is incarceration defined?

Response: We have now clarified under section 2.2. Independent Variable that incarceration exposure is self-reported and measured using a survey item that asks women whether in the 12 months before the current birth “I or my husband/partner went to jail”

  1. Is the length of incarceration known? If yes, does this make a difference? An unknown length of incarceration should be addressed in the limitations section.

Response: Unfortunately, the PRAMS survey does not capture the length of incarceration. We have now addressed this in the limitation section. Even so, we also note that prior research on the consequences of incarceration for health tend to find that the effects of the length of incarceration on health outcomes are less important than incarceration itself (Massoglia 2008; Schnittker and John 2007). Still, we note that future research that explores the impact of incarceration length on barriers to prenatal care would be useful.

Massoglia, M. (2008). Incarceration as exposure: the prison, infectious disease, and other stress-related illnesses. Journal of Health and Social Behavior49(1), 56-71.

Schnittker, J., & John, A. (2007). Enduring stigma: the long-term effects of incarceration on health. Journal of Health and Social Behavior48(2), 115-130.

  1. In the year before the birth suggestions that the woman/partner could have been incarcerated and then released prior to the woman becoming pregnant. Is the timing of the pregnancy known? If yes, does when during the pregnancy (before, 1st, 2nd, 3rdtrimester) make a difference? An unknown timing of pregnancy should be addressed in the limitations section.

Response: The reviewer is correct in pointing out a limitation of the measure. Specifically, based on the wording of the question, we only know whether an incarceration event took place at some point in the 12 months prior to birth. Therefore, it is possible that for some, a person could have been incarcerated shortly prior to, or during pregnancy, but released during birth. Unfortunately, this is a limitation of the PRAMS study, which we have now explicitly detailed in the limitations section.

  1. If both length and timing of incarceration known, does their combined potential effect make a difference.

Response: As we noted above, neither timing nor length of incarceration is known. However, we have expanded the limitations section to encourage future research to explore this possibility with alternative sources of data.

  1. Who is incarcerated? While the authors address this issue in the limitations section, I feel that it is not given enough attention. Can the authors talk about any potential benefits that a young, non-white, poorly educated, woman from a low SES neighborhood might obtain in terms of access to prenatal care if she is incarcerated? I stipulate this population as it describes the present sample. However, can being incarcerated potentially benefit pregnant women with lower resources and means?

Response: While we do not know exactly who is incarcerated, we have added some statistics suggesting that in recent years over 93% of people incarcerated in prison and 85% of those incarcerated in jail are males. Therefore, we can approximate that for most of the sample, the person who experienced the incarceration is the husband/partner of the pregnant woman, rather than the woman herself. Even so, we have now added a part to the discussion talking about how incarceration may provide benefits for prenatal care access to certain disadvantaged segments of the population – namely women with low resources and means.

  1. The authors make a few suggestions (taxi vouchers, public assistance, vouchers for childcare) that might mitigate the effects of incarceration exposure on accessing prenatal care. Have these factors been useful in mitigating prenatal care in other disadvantaged populations? This topic needs to be better developed as to why the authors think they may be beneficial for the population under study above and beyond that observed in other disadvantaged populations.

Response: We thank the author for this helpful comment. To be sure, there has been somewhat limited research on the efficacy of interventions to overcome barrier to prenatal care among disadvantaged populations. Still, we have reviewed the literature and suggest interventions that are specifically applicable to this population. In doing so, we have revised the manuscript to note that these interventions have shown effectiveness among disadvantaged populations.

  1. In the Discussion the author write: “As such, this may result in a reduction in ownership of household assets such as a vehicle [31], as well as exacerbate financial hardship, thereby making public transportation costlier and more difficult to obtain.”

Response: We have now clarified this statement to note that we are referring to prior research that explicitly links incarceration to reduces vehicle ownership and financial hardship as a mechanism that might explain a higher rate of incarceration exposed women noting transportation issues as a barrier to prenatal care.

  1. While semantic in nature, it is incorrect to conclude that incarceration exposure is associated with costlier and more difficult to obtain public transportation. The cost and frequency of public transportation is independent of the women’s incarceration status. Incarceration exposed women may experience greater financial difficulties which in turn may make it financially difficult to use public transportation, but it does not make public transportation costlier.

Response: We have now revised this statement to clarify incarceration exposed women may experience greater financial difficulties which in turn may make it financially difficult to use public transportation, but it does not make public transportation costlier. We follow this statement up with a suggestion that providing vouchers to various public transportation options can be an effective means of improving access to prenatal care for incarceration-exposed women.

  1. How was access to early childcare defined?

Response: We have now clarified in the manuscript that this does not refer to “early childcare.” Instead, this refers to a lack of having someone to care for children, which inhibited women from being able to obtain prenatal care appointments when they desired. Specifically, we report in appendix A that women answered in the affirmative to the following statement “I had no one to take care of my children.”

  1. Figure 1. It would be useful if the authors identify which barriers differ significantly between the 2 groups.

Response: We have now altered figure 1 to indicate which barriers significantly differ between the two groups.

  1. Why is the sample size for Figure 1 (N = 34,658) different from the sample listed for Table 1 (N = 194,600)?

Response: We have now clarified in the manuscript under section 2.4. Analytic Approach that the sample 194,600 is the full sample of women with observed data on the variables included in the analysis. The sample 34,658 is a subsample of women who reported at least one barrier to prenatal care. Therefore, the difference of these samples of 159,942 (194,600 – 34,658) is the number of women who reported having no difficulty accessing prenatal care.

  1. Why are “Didn’t want PNC” and “Didn’t know pregnant” listed as barriers to prenatal care? Being too busy, not having childcare, time of work, enough money, a Medicaid care, … cannot be seen as barriers if the person has either stipulated they do not wish to seek prenatal care or are do not know that they require medical care. Please clarify.

Response: We thank the reviewer for pointing this out and agree that some of the barriers refer more to structural barriers, whereas others refer to cognitive or knowledge-based barriers. Ultimately, we have opted to include all 11 barriers in the measure for the following reasons. First, the 11-barrier questions are asked together as part of the PRAMS Phase 6 survey (see section 27 [p. 111] https://www.cdc.gov/prams/pdf/questionnaire/Phase6_TopicsReference.pdf). Accordingly, we felt that including all the barriers was consistent with the intent of the survey designers. Second, even though the latter two-barriers (Didn’t want PNC; Didn’t know pregnant) are not structural barriers to PNC, we argue that they still remain important barriers given the wide-ranging benefits of PNC, and therefore exploring these outcomes may still yield important information and knowledge regarding why PNC access among incarceration-exposed women is lower than non-incarceration exposed women. Third, we also now note under section 2.1 Dependent Variable that these 11 items encompass a wide range of barriers broadly defined in the extant literature, which encompass structural, cognitive, and knowledge-based issues that may inhibit women from getting adequate prenatal care (Braveman et al., 2000; Campbell et al., 1996; Phillippi et al., 2009). Finally, as we mention in the comment below, we have re-estimated our analysis without these two barriers and find the results remain substantively similar.

Braveman, P., Marchi, K., Egerter, S., Pearl, M., & Neuhaus, J. (2000). Barriers to timely prenatal care among women with insurance: the importance of prepregnancy factors. Obstetrics & Gynecology95(6), 874-880.

Campbell, J. D., Stanford, J. B., & Ewigman, B. (1996). The social pregnancy interaction model: Conceptualizing cognitive, social and cultural barriers to prenatal care. Applied Behavioral Science Review4(1), 81-97.

Phillippi, J. C. (2009). Women's perceptions of access to prenatal care in the United States: a literature review. Journal of Midwifery & Women's Health54(3), 219-225.

  1. Do the results of the initial analysis (number of barriers) change if these two “barriers” are removed from the total score of barriers?

Response: We have reproduced the initial analysis, which is reported as Table S1 at the bottom of this memo. As the results show, the findings remain substantively similar if these two barriers (didn’t want PNC and didn’t know pregnant) are removed.

  1. Figure 2. The labelling for “Didn’t want PNC appears to be incorrect. The bar graph suggests an OR of greater than 1.40.

Response: We thank the author for catching this error and have now corrected the coefficient to 1.36. The bar is now below the 1.40 line.

  1. Net of covariates. This is a phrase that I have not come across before. I asked three statisticians how they would interpret the following sentence that authors wrote I the Results section (“To do so, we conduct separate logistic regression models using responses to each of the 11 barriers to prenatal care as the outcome variable, net of covariates.") and received three different responses. One believed that the phrase suggested that the covariates were included in the analyses, one believed that the covariates were excluded from the analyses, and the third did not know whether the covariates were or were not included. I ask for clarification since my interpretation is without covariates. If my interpretation is true, then the text appears to report OR without the influence of the covariates while Figure 2 appears to report ORs after controlling for the covariates.

Response: We have now revised the sentence to state “To do so, we conduct separate logistic regression models using responses to each of the 11 barriers to prenatal care as the outcome variable while controlling for covariates.”

Table S1. Results of Negative Binomial Regression of Incarceration Exposure on Number of Prenatal Care Barriers (N = 194,600).

VARIABLES

Model 1

Model 2

IRR

95% CI

IRR

95% CI

Incarceration Exposure

2.459***

(2.229 - 2.714)

1.558***

(1.406 - 1.726)

Maternal Age

17 or Younger (Reference)

-

-

-

-

18-24

0.964

(0.843 - 1.103)

25-29

0.849*

(0.736 - 0.980)

30-34

0.743***

(0.638 - 0.865)

35+

0.712***

(0.605 - 0.838)

Maternal Race/Ethnicity

White (Reference)

-

-

-

-

Hispanic

1.170***

(1.084 - 1.264)

Black

1.071+

(0.995 - 1.153)

Other Race/Ethnicity

1.501***

(1.378 - 1.634)

College Graduate

0.896**

(0.828 - 0.971)

Currently Married

0.871***

(0.816 - 0.930)

Number of Prior Births

0 (Reference)

-

-

-

-

1

0.998

(0.938 - 1.063)

2

1.062

(0.984 - 1.147)

3+

1.364***

(1.239 - 1.503)

Pregnancy Planned

0.591***

(0.557 - 0.626)

Income Levels

Less than $10,000 (Reference)

-

-

-

-

$10,000-14,999

1.022

(0.920 - 1.136)

$15,000-19,999

0.910

(0.801 - 1.032)

$20,000-24,999

0.835***

(0.766 - 0.912)

$25,000-34,999

0.726***

(0.660 - 0.798)

$35,000-$49,999

0.596***

(0.535 - 0.664)

$50,000 or Greater

0.301***

(0.272 - 0.334)

Constant

0.453***

(0.393 - 0.523)

0.828+

(0.675 - 1.014)

State Dummy Variables

Yes

Yes

Year Dummy Variables

Yes

Yes

        *** p<.001, ** p<.01, * p<.05, † p<.10.

Reviewer 3 Report

  1. This article is a useful investigation.
  2. However, at least some of its conclusions are almost expected from reading its title.
  3. The article lacks originality. At least in some parts, it seems a repeat of already published research.
  4. The article is worth to be published. However, it would be useful if some suggestions to mentioned barriers could be presented at the end of the discussion.

Author Response

  1. This article is a useful investigation.

Response: We thank the reviewer for the positive assessment of the manuscript.

  1. However, at least some of its conclusions are almost expected from reading its title.

Response: We aimed to create a title to be descriptive of the findings to guide readers. We are happy to change the title at the editor’s request.

  1. The article lacks originality. At least in some parts, it seems a repeat of already published research.

Response: We have now revised the manuscript, especially the introduction to highlight the novelty of the current study.

  1. The article is worth to be published. However, it would be useful if some suggestions to mentioned barriers could be presented at the end of the discussion.

Response: We have now revised the end of the discussion to provide additional suggestions to overcoming some of the key barriers.
